# Mechanical Properties of MiniBars™ Basalt Fiber-Reinforced Geopolymer Composites

**DOI:** 10.3390/ma17010248

**Published:** 2024-01-02

**Authors:** Gabriel Furtos, Doina Prodan, Codruta Sarosi, Marioara Moldovan, Kinga Korniejenko, Leonard Miller, Lukáš Fiala, Nováková Iveta

**Affiliations:** 1Raluca Ripan Institute of Research in Chemistry, Babes-Bolyai University, 400294 Cluj-Napoca, Romania; doina.prodan@ubbcluj.ro (D.P.); codruta.sarosi@gmail.com (C.S.); mmarioara2004@yahoo.com (M.M.); 2Faculty of Materials Engineering and Physics, Cracow University of Technology, 31-864 Cracow, Poland; kinga.korniejenko@pk.edu.pl; 3ReforceTech AS, NO-3440 Røyken, Norway; len.miller@reforcetech.com; 4Department of Materials Engineering and Chemistry, Faculty of Civil Engineering, Czech Technical University in Prague, 166 29 Prague, Czech Republic; lukas.fiala@cvut.cz; 5Faculty of Science and Technology, The Arctic University of Norway, N-8505 Narvik, Norway; iveta.novakova@uit.no

**Keywords:** MiniBars, basalt fiber, fly ash, geopolymer composites, mechanical properties

## Abstract

Fly ash-based geopolymers represent a new material, which can be considered an alternative to ordinary Portland cement. MiniBars™ are basalt fiber composites, and they were used to reinforce the geopolymer matrix for the creation of unidirectional MiniBars™ reinforced geopolymer composites (MiniBars™ FRBCs). New materials were obtained by incorporating variable amount of MiniBars™ (0, 12.5, 25, 50, 75 vol.% MiniBars™) in the geopolymer matrix. Geopolymers were prepared by mixing fly ash powder with Na_2_SiO_3_ and NaOH as alkaline activators. MiniBars™ FRBCs were cured at 70 °C for 48 h and tested for different mechanical properties. Optical microscopy and SEM were employed to investigate the fillers and MiniBars™ FRBC. MiniBars™ FRBC showed increasing mechanical properties by an increased addition of MiniBars™. The mechanical properties of MiniBars™ FRBC increased more than the geopolymer wtihout MiniBars™: the flexural strength > 11.59–25.97 times, the flexural modulus > 3.33–5.92 times, the tensile strength > 3.50–8.03 times, the tensile modulus > 1.12–1.30 times, and the force load at upper yield tensile strength > 4.18–7.27 times. SEM and optical microscopy analyses were performed on the fractured surface and section of MiniBars™ FRBC and confirmed a good geopolymer network around MiniBars™. Based on our results, MiniBars™ FRBC could be a very promising green material for buildings.

## 1. Introduction

Portland cement is one of the basic components for obtaining concrete, but the main problem is the CO_2_ emissions. Despite its high compressive strength and durability, concrete’s use is limited by its low tensile strength, crack propagation, and the disadvantage that it has a major role in global warming [1]. The cement industry, as a whole, had a contribution of around 7–8% to CO_2_ emissions worldwide [2]. Due to these reasons, alternative methods are constantly being searched for, in order to obtain more ecological materials, able to replace Portland cement totally or partially.

Fly ash, as waste resulting from the burning of coal, also represents a serious problem for the environment. Fly ash could be used in the development of geopolymer concrete as a green building material, a very cheap alternative to Portland cement, that contributes to the circular economy [3,4,5,6]. Geopolymer concretes are prepared using the precursors of slag, fly ash, and the activators of NaOH, KOH, etc., and water glass (Na_3_SiO_3_). At basic attack of the activator, a dissolution of aluminosilicate leads to the forming of new bonds, and an amorphous three-dimensional geopolymer matrix will be developed, after a polycondensation reaction and bonding of inorganic fillers influenced by curing at ambient or high temperatures [7,8].

Therefore, geopolymer concrete showed the same or even improved properties compared to ordinary Portland concrete: early compressive strength, low permeability, good chemical resistance, excellent fire resistance and lower CO_2_ emissions [3,9]. In order to improve the mechanical properties of geopolymers, glass/basalt fibers were used for concrete reinforcement; recently, MiniBars basalt fibers were used for this purpose. MiniBars™ is a high-performance basalt fiber composite that can provide advantages to concrete: it improves the post-cracking mechanical properties of hardened concrete, increasing the toughness and impact and fatigue resistance of concrete, and does not corrode [10].

Also, the fibers can be impregnated with an alkali-resistant polymer resin, and the matrix–fiber connection can be mechanical or chemical–mechanical; a chemical bond between the matrix and the fiber would increase the resistance to friction.

Due to the fact that they are resistant to alkalis and have high strength and rigidity, basalt fibers can significantly increase the strength and ductility of concrete. Degradation of glass fibers in concrete can be stopped by using fibers of a suitable size, by adjusting the concrete binder or by impregnating the fibers with polymer, and the corrosion will be less than that of steel [11]. Basalt fibers are readily available and have a relatively low price. Under the action of the load force on the cement matrix, microcracks could be reduced or stopped and the flexural resistance can be improved [12] The addition of glass fiber to reinforce geopolymer concrete increases tensile strength by 5–10% [13]. The performance of fiber-reinforced materials mainly depends on the fiber content, their arrangement, their type, dimensions, and their compatibility with the cement matrix [11,14].

The aim of this study was to develop a new MiniBars™ basalt fiber-reinforced geopolymer composite (MiniBars™ FRBC). The fly ash morphology, sizes and structure were determined by optical microscopy and scanning electron microscopy (SEM). New MiniBars™ FRBCs were investigated for structure through optical microscopy and SEM. Also, the flexural strength, the flexural modulus, the tensile strength, the tensile modulus, and the force load at upper yield tensile strength were analyzed.

## 2. Materials and Methods

### 2.1. Materials

MiniBars™ FRBCs were prepared using class-F fly ash (Figure 1a) from a coal power plant (Mintia, Romania), sodium silicate (Na_2_SiO_3_) and sodium hydroxide (NaOH), which were acquired from AGEXIM SRL, Romania. MiniBars™ (Figure 1b) is a fiber-reinforced polymer (FRP) based on basalt fibers (helix roving with 1200 basalt fibers, with diameter of basalt fiber of 17 μm). MiniBars™ had a 55 mm length and the fibers were coated with thermoset resin (vinyl ester resin) and the diameter of the MiniBars™ reached 0.70 mm. MiniBars™ were received from ReforceTech AS, Røyken, Norway. According to ReforceTech AS Company, the MiniBars™ contain 80% basalt fiber and 20% resin, and their density is around 2.1 g/cm^3^. The degree of crystallinity for fly ash and geopolymer from X-ray diffraction from previous work [6] was calculated as the ratio between the area of the diffraction peaks due to the crystalline phases, divided by the sum of the diffraction areas due to the crystalline phases, plus the areas coming from the halos of the amorphous phases, using Equation (1) [15]: D_cr_ = (I_c_)/(I_c_ + I_a_)(1)
where D_cr_ is the degree of crystallinity, I_c_ is the sum of the areas from the crystalline phases in the sample and I_a_ is the area due to the halos from the amorphous phases.

Fly ash samples were characterized in previous work [6] using a spectrofluorometer (JASCO FP-6500, Tokyo, Japan) for the composition and the size distribution, which was 0.103 μm at Dv50. According to those measurements, SiO_2_/Al_2_O_3_ ratio from fly ash was 3.07, and it was confirmed being in Class F of fly ash. A fresh solution of the NaOH 14 M was prepared and kept cooling down to room temperature. A sodium silicate with modulus SiO_2_/NaOH = 2.5 was mixed with sodium hydroxide solution at a ratio of Na_3_SiO_3_/NaOH = 2.5:1. The alkali hydroxide solution was mixed with fly ash in order to dissolve the surface of aluminosilicate from fly ash, while the Na_2_SiO_3_ solution was used as a binder for all the obtained small cations.

### 2.2. Preparation of MiniBars™ FRBC

For preparing the samples, we used a rectangular mold (20 mm ± 0.1 × 20 mm ± 0.1 × 70 mm ± 0.1). Before, to propose the composition, we filled all the volume of the mold with MiniBars™ across the length of mold and measured the weight of these fibers (19.50 g = 100% vol). Based on this weight, we proposed for investigation MiniBars™ FRBC with 0; 12.5; 25; 50; and 75 vol.% MiniBars™. The mixture proportions of the prepared MiniBars™ FRBC (*n* = 8) are shown in Table 1. The geopolymer paste was obtained by mixing the fly ash with the liquid Na_3_SiO_3_/NaOH at a weight ratio of 1.67. When the homogeneity of geopolymer paste was achieved, we filled half of the mold with the geopolymer paste and then added the MiniBars™ in the geopolymer paste. MiniBars™ were moved down, up and sideways with a stick in the paste to be penetrated and moistened by the geopolymer paste. The mold was overfilled by the geopolymer paste. In order to remove entrapped air from the samples, the mold was kept for 5 min on a vibrating table. The samples (*n* = 8–10) were covered with plastic film and cured at 70 °C for 48 h. After curing, the specimens were removed from the mold, and those with voids were excluded from this investigation.

### 2.3. Mechanical Characterization of MiniBars™ FRBC

All mechanical tests (*n* = 8–10) were carried out using a universal testing machine (LR5K Plus, Lloyd instruments, Ltd., London, UK) at a loading rate of 1 mm/min^−1^ until fracture. The load–deflection curves were recorded using computer software (Nexygen; Ver. 4 Lloyd Instruments, UK).

The flexural strength (FS) was calculated using Equation (2).
FS = 3F_max_l/2bh^2^(2)
where FS is the flexural strength (MPa), F_max_ is the applied load (N), l is the span between the supports (50 mm), b is the width (20 mm), and h equals the thickness (20 mm). The flexural modulus (MPa) was determined from the slope of the elastic portion of the stress–strain curve.

The tensile strength (TS) was measured using Equation (3).
(3)$$TS=\frac{F}{A}(MPa)$$
where TS is the tensile strength (MPa), F is the force on the cross-section of the specimen at ultimate tension (N) and A is the nominal cross-sectional area of the specimen (mm^2^). The tensile modulus (MPa) was determined from the slope in the elastic portion of the stress–strain curve. The force load at upper yield tensile strength (KN) was recorded in order to determine the maximum force in the elastic area for the MiniBars™ FRBC.

### 2.4. Scanning Electron Microscopy (SEM) and Optical Microscopy

The sample morphology of the fly ash, MiniBars™, MiniBars™ FRBC and the structure of the fractured surfaces of MiniBars™ FRBC specimens were investigated using a stereomicroscope (Stemi 2000-C, Carl Zeiss AG, Oberkochen, Germany), as well as SEM (SEM Inspect S, FEI, Eindhoven, The Netherlands). The fly ash, the wood fibers and the surfaces of MiniBars™ FRBC specimens after the FS test were also evaluated using a stereomicroscope (Stemi 2000-C, Carl Zeiss AG).

## 3. Results and Discussion

The degree of crystallinity was determined as being 40.1% for fly ash and 36.2% for the geopolymer. The white spherically shaped particles could be Mulite (Figure 2a, yellow arrow) [16] and the black particles Hematite—Fe_2_O_3_ (Figure 2a, red arrow). These glass or vitreous fragments and spherical particles were observed also by SEM (Figure 2b yellow arrow) and were mentioned also in other articles [6,17]. Our results from previous XRD analysis showed that the detectable phases in the fly ash were Quartz—SiO_2_ (PDF#461045), Mullite—3Al_2_O_3_·SiO_2_ (PDF#150776), and Hematite—Fe_2_O_3_ (PDF#330664) in a small amount [6]. In the used fly ash, the vitreous phase (spherical or nonregular shape) exists together with the crystalline one [6].

In Figure 3, we can see unreacted spherical particles (yellow arrow) of fly ash of different sizes covered and interconnected by the geopolymer matrix (red arrow) that will contribute to the improvement of mechanical properties. These spherical particles were also observed above in Figure 2 by optical microscopy and SEM. This behavior was in agreement with other publications [5,6,17,18].

All mechanical test values increased (Figure 4) with the addition of MiniBars™ and were more improved than Fly100. The highest value was for MiniBars75. Flexural strength of MiniBars™ FRBCs (Figure 4a) increased with the addition of MiniBars™ and increased more than 11.59–25.97 times than Fly100 (geopolymer without fibers) The values registered for the flexural strength of MiniBars™ FRBCs were between 9.99 and 22.39 MPa. The flexural modulus of MiniBars™ FRBCs (Figure 4b) increased in the same way with the increased addition of MiniBars™: more than 3.33–5.92 times than that of Fly100, with values between 267.74–475.63 MPa. These results were in agreement with other studies when the addition of glass fiber increased mechanical properties and fibers acted as crack stoppers [19,20,21,22]. Under the load applied to MiniBars™ FRBC, the cracks will be bridged by MiniBars™. Flexural strength and flexural modulus of MiniBars™ were well-correlated with the quantity of MiniBars™ (R^2^ = 0.9458 and R^2^ = 0.945). The continuous unidirectional fibers showed anisotropic mechanical properties, and the highest strength and stiffness were obtained when the direction of the force load was the same as the orientation of the fibers [23]. Krenchel’s reinforcing factor [24] showed that the 3D randomly oriented short fibers (chopped fibers) in composites gave a strengthening factor of 0.2, whereas 2D-oriented fibers (woven) gave 0.375 and unidirectional fibers gave a factor of 1.

The tensile strength of MiniBars™ FRBCs (Figure 4c) increased with the addition of MiniBars™ between 3.08–7.06 MPa. The highest values were registered for MiniBars75. The results obtained were between 3.50–8.03 times higher than Fly100. The results of tensile strength for MiniBars™ FRBCs were well-correlated (R^2^ = 0.9665) with the quantity of MiniBars™. Tensile modulus of MiniBars™ FRBCs (Figure 4d) increased with the addition of MiniBars™ more than 1.12–1.30 times than that of Fly100 and was less than the tensile strength. The tensile modulus values were between 292.58 and 341.68 MPa. Tensile modulus of MiniBars™ FRBCs correlated with the quantity of MiniBars™ and showed very good regression (R^2^ = 0.9542).

The force load at upper yield tensile strength of MiniBars™ FRBCs (Figure 4e) showed values of 4.18–7.27 times more than Fly100. Force loads at upper yield of MiniBars™ FRBCs were between 1.23 and 2.14 KN and the results showed that MiniBars™ loads were well-correlated with the quantity of MiniBars™ (R^2^ = 0.9205).

In the photographs of the sections of sample MiniBars™ FRBCs (Figure 5a–d) and the optical images of sections of MiniBars™ FRBCs (Figure 5e–h), we can see MiniBars™ distribution in the transverse section of the samples: black circle (yellow arrow) and around these MiniBars™ geopolymer (red arrow). These images present an image of the distribution of the MiniBars™ in the mold. The distribution was made in the rectangular mold by a stick, which moved the MiniBars™ down, up and sideways in the paste, which vibrated 5 min on a vibrating table. During last step on the vibrating table, the MiniBars™ will be rearranged in the mold volume depending on the quantity of MiniBars™. In Figure 5, we can see MiniBars™ dispersed in all the volume of the samples because there was more free space. Figure 5b shows more MiniBars™ in half of the samples, which could be explained by the fact that MiniBars™ under vibration made a rearrangement and fibers were placed on top of each other. In Figure 5c,d, MiniBars™ try to fill all the volume of the samples and look more uniformly dispersed. The limitation of this research lies in the fact that it is impossible to see what happens after we cover the MiniBars™ with geopolymer pastes, and we control them only with a stick in the paste, and in the end the vibration process decides the arrangement of the fibers in these samples.

Figure 6 show how MiniBars™ stop the crack propagation in the geopolymer matrix of MiniBars™ FRBCs under load force and materials display elastic behavior (Figure 6d). When the force load increases, the geopolymer matrix will be the first that will break and not the MiniBars™. The fracture will be generated in the direction of the load force. The geopolymer matrix will break from outside (contact of load force) to inside by debonding from the surface of the MiniBars™. During the test, the geopolymer from around the MiniBars™ will break but not the fibers, and they will show an elastic behavior, a property that is not available to geopolymer concrete. During the test, even the geopolymer matrix starts to break on the direction of the force load, but left and right of this, MiniBars™ will try to keep the geopolymer bonded to the surface of MiniBars™ until the samples fail at mechanical testing.

These results were in agreement with other studies regarding the stopping of crack propagation of glass/basalt fiber in composites [20,21,22]. In our study, unidirectional MiniBars™ showed higher mechanical properties than Fly100. This could be explained by the direction of orientation of MiniBars™ (unidirectional) that was perpendicular to the direction of the applied force [21]. The strength of the MiniBars™ FRBC also depends on the fiber direction inside composites [21]. In our case, when the matrix crack grows at 90 degrees of unidirectional fiber alignment, the fibers or MiniBars™ will form a fiber-bridged crack. The continuous unidirectional fibers from MiniBars™ will provide the anisotropic mechanical properties, with the highest strength and stiffness when the direction of the applied load force and the orientation of the fibers is the same. This was in agreement with another study [21].

Figure 7a shows SEM images of MiniBars™ that did not have a regular shape because during the curing some of the basalt fibers, roving, absorbed more resin. In Figure 7b–d, we can see sections of MiniBars™ (yellow arrow), geopolymer with good adhesion around MiniBars™ (red arrow), basalt fiber from roving or MiniBars™ (green arrow), resin around basalt fiber from roving or MiniBars™ (blue arrow). Figure 7e,f show the adhesion of geopolymer on basalt fiber from MiniBars™ after a flexural test. The light blue arrow indicates some holes at the surface of fibers where geopolymer was pulled out or fractured. This is the mechanical interlock effect on adhesion strength of geopolymer–fiber interfaces, and it will contribute to the adhesion of fibers to the geopolymer matrix. Even though there is not a chemical bond between the geopolymer matrix and resin at the surface of the fiber, the mechanical interlock effect from the geopolymer at the surface of the fibers and between MiniBars™ will contribute to the increase in mechanical properties.

The advantages of basalt fibers was mentioned recently in another study [22], where the shear resistance increased by 95%, 98%, 136% and 210% and the post-cracking stiffness of all beams showed a remarkable increase. The compressive strength and modulus of rupture of basalt fiber-reinforced concrete increase with the addition of basalt fiber and early shrinkage cracks decrease with an increase in the basalt fiber volume fraction [25].

## 4. Conclusions

All mechanical properties increase when the addition of MiniBars™ increases in MiniBars™ FRBCs. The best mechanical properties were obtained for MiniBars75. The mechanical properties of MiniBars™ FRBCs increased more times than those of the geopolymer without MiniBars™: the flexural strength > 11.59–25.97 times, the flexural modulus > 3.33–5.92 times, the tensile strength > 3.50–8.03 times, the tensile modulus > 1.12–1.30 times, and the force load at upper yield tensile strength > 4.18–7.27 times. The addition of MiniBars™ stopped the crack propagation of the geopolymer matrix and increased the mechanical properties. The SEM and optical microscopy confirmed a good geopolymer network around MiniBars™ in the MiniBars™ FRBCs. All these results confirm that MiniBars™ FRBCs could be a very promising green material for buildings and MiniBars™ could replace steel rods for reinforcing geopolymers.

## Figures and Tables

**Figure 1 materials-17-00248-f001:**
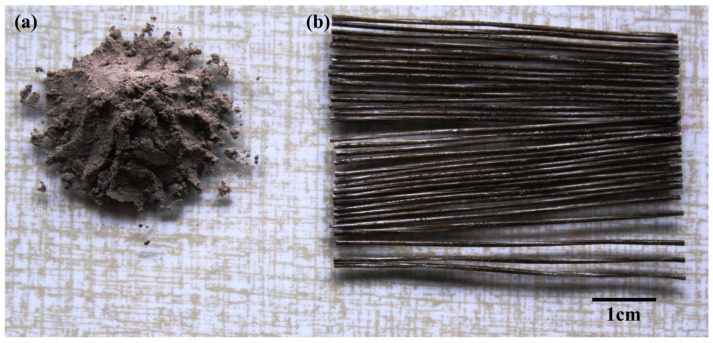
The photographs of: (**a**) fly ash powder; (**b**) MiniBars™.

**Figure 2 materials-17-00248-f002:**
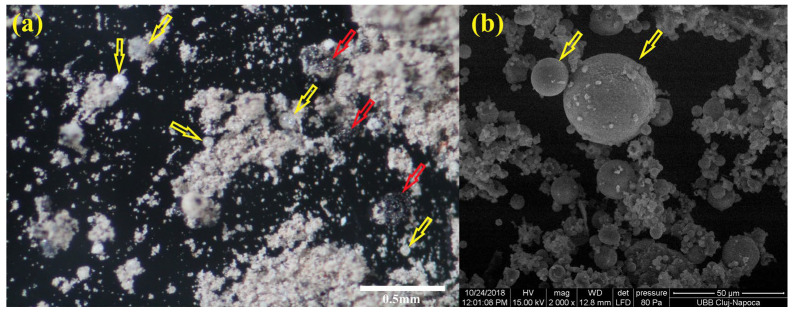
(**a**) Optical microscopy of fly ash powder; (**b**) SEM micrographs of fly ash powder.

**Figure 3 materials-17-00248-f003:**
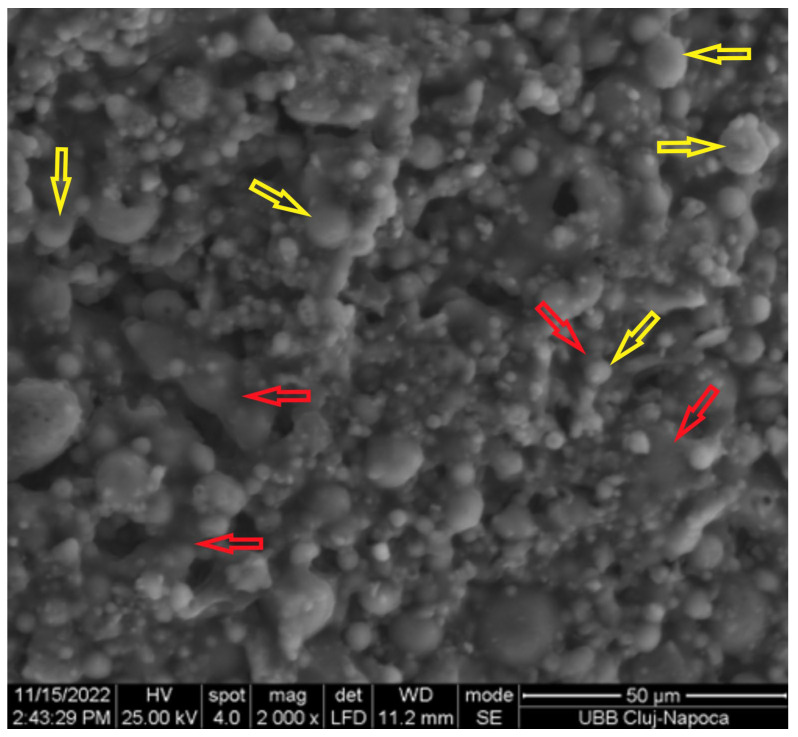
SEM micrographs of the surface of cured geopolymer.

**Figure 4 materials-17-00248-f004:**
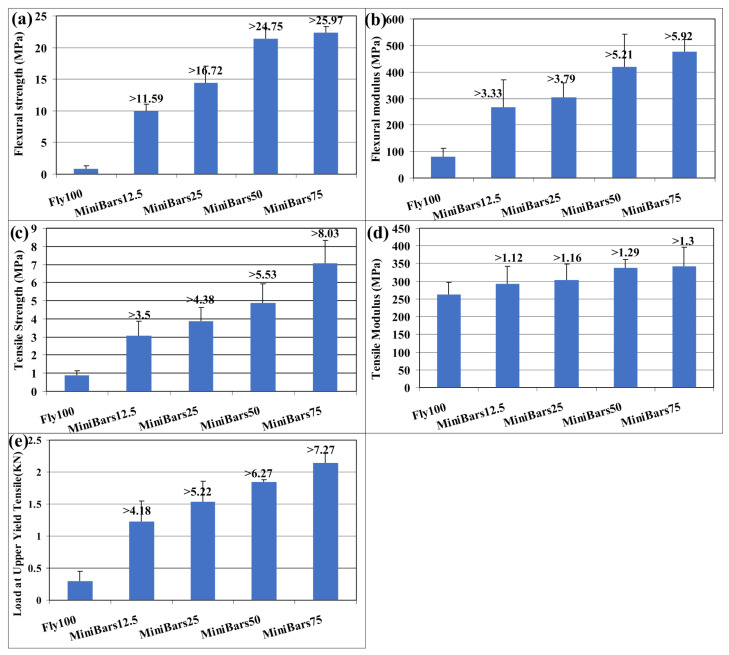
Mechanical properties of MiniBars™ FRBCs: (**a**) flexural strength; (**b**) flexural modulus; (**c**) tensile strength; (**d**) tensile modulus; (**e**) force load at upper yield tensile strength.

**Figure 5 materials-17-00248-f005:**
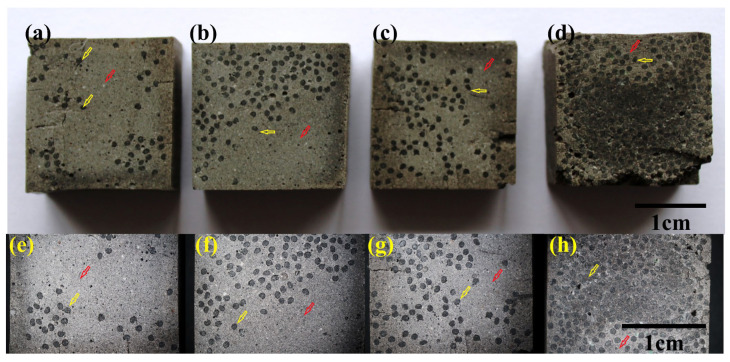
The photographs of the sections of samples of MiniBars™ FRBCs: (**a**) MiniBars12.5; (**b**) MiniBars25; (**c**) MiniBars50; and (**d**) MiniBars75. Optical images of the sections of MiniBars™ FRBCs: (**e**) MiniBars12.5; (**f**) MiniBars25; (**g**) MiniBars50; and (**h**) MiniBars75.

**Figure 6 materials-17-00248-f006:**
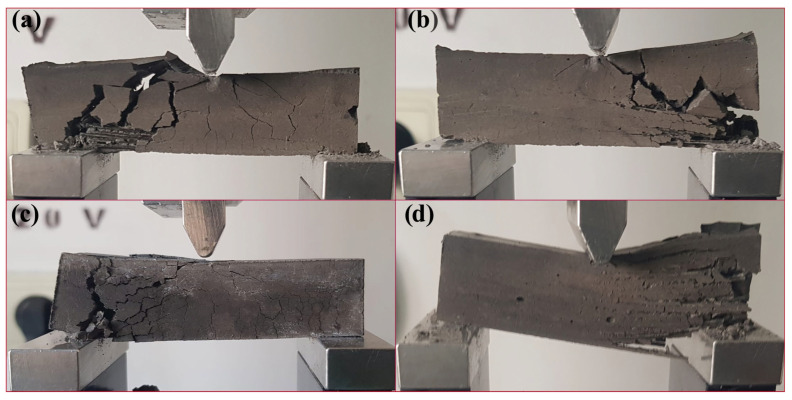
The photographs of fractured MiniBars™ FRBCs at FS test: (**a**) MiniBars12.5; (**b**) MiniBars25; (**c**) MiniBars50; and (**d**) MiniBars75.

**Figure 7 materials-17-00248-f007:**
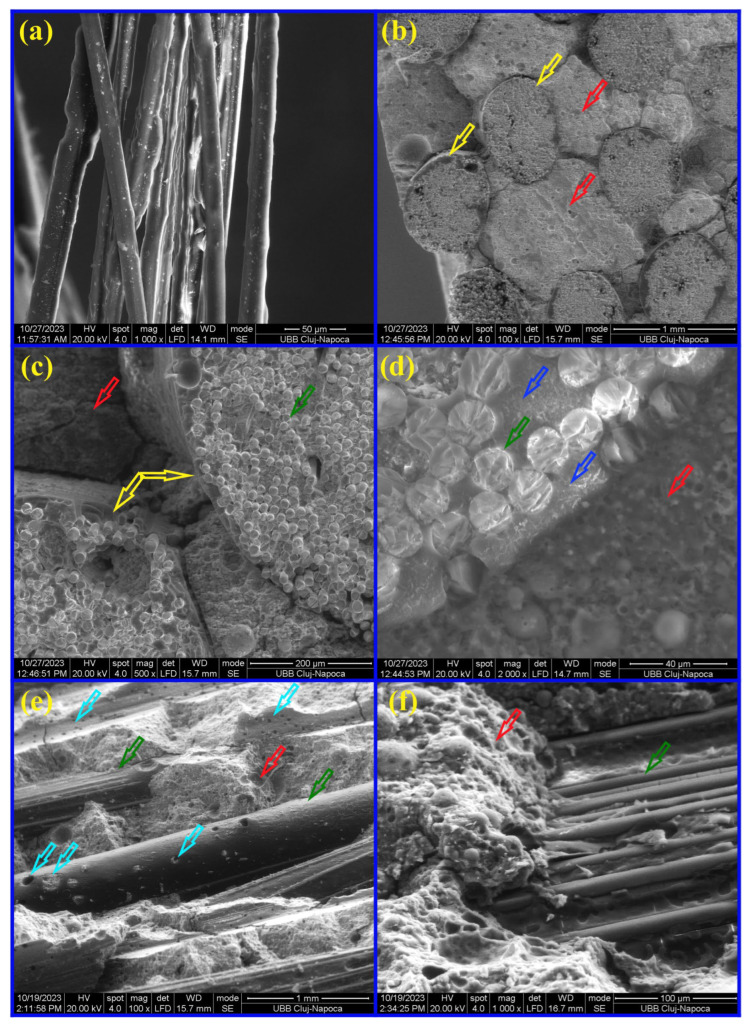
SEM images of: (**a**) MiniBars™; (**b**–**d**) transverse section of MiniBars75 after flexural test; (**e**,**f**) fracture of MiniBars75 after flexural test.

**Table 1 materials-17-00248-t001:** Composition of the MiniBars™ FRBCs.

Nr.	Code	vol. % MiniBars	wt. % MiniBars
1	Fly100	0	0
2	MiniBars12.5	12.5	2.44
3	MiniBars25	25	4.88
4	MiniBars50	50	9.75
5	MiniBars75	75	14.63

Note: 100 vol.% MiniBars in rectangular mold (20 mm × 20 mm × 70 mm) had the weight of 19.50 g.

## Data Availability

The data that support the findings of this study are contained within the article.

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
