# Peer review of "Mechanical Properties of MiniBars™ Basalt Fiber-Reinforced Geopolymer Composites"

_materials, 2024, doi:10.3390/ma17010248_

Round 1

Reviewer 1 Report

Comments and Suggestions for Authors

The article is devoted to the study of obtaining a new MiniBars™ basalt fiber reinforced geopolymer composites. The authors investigate mechanical characterization (flexural strength, the flexural modulus, the tensile strength, the tensile modulus, the force load at upper yield tensile strength) of  New MiniBars™ 85 FRBC. Characterization techniques such as optical microscopy, scanning electron microscopy (SEM) were used.

At the same time, there are specific comments to this article:

Lines 57, 105, 116: the formula of water glass (Na3SiO4) is given. In this formula, the balance of valences does not converge.

Line 97: 0.70 mm diameter of MiniBars™. At the same time, the length should be specified.

Line 100: Fly-ash samples were characterized in work [6] using a spectrofluorometer (JASCO FP-6500, Japan) for the composition and the size distribution, which was 0.103 μm at Dv50. It is also necessary to indicate the specific surface area of fly ash.

Lines167-170: It is indicated that the phases that can be detected in fly ash are: quartz - SiO2 (PDF#461045), mullite - 3Al2O3·SiO2 (PDF#150776) and hematite - Fe2O3 (PDF#330664) in small quantities. In the used fly ash, the vitreous phase (spherical or irregular) exists  together with the crystalline phase [6].

It is also necessary to present the quantitative content of these crystalline phases, as well as the content of the vitreous phase. It is also necessary to give the equation for the interaction of liquid glass with fly ash components and to determine which interaction products are formed and to present their stoichiometry

Line 183: All mechanical testing values increase (Figures 4) with addition of MiniBars™  and were more improved than Fly100.  It is necessary to provide compressive strength at cubic test for the tested samples of Fly 100.

Figure 5. The photographs of the section of samples of MiniBars™ FRBC: a) MiniBars12.5; b) 221 MiniBars25; c) MiniBars50 and d) MiniBars75.  This shows the uneven distribution of fiber over the area of samples. Therefore, the anisotropy of mechanical characteristics should be manifested. However, this factor is not taken into account in the article.

Author Response

Dear reviewer 1,

I would like to thank you very much for you time spent to read our paper.

We try to improve the article by some changes in the text. The new changes were colored yellow highlight.

Thank you very much for all your help. Please see below my answer to your suggestions:

The article is devoted to the study of obtaining a new MiniBars™ basalt fiber reinforced geopolymer composites. The authors investigate mechanical characterization (flexural strength, the flexural modulus, the tensile strength, the tensile modulus, the force load at upper yield tensile strength) of  New MiniBars™ 85 FRBC. Characterization techniques such as optical microscopy, scanning electron microscopy (SEM) were used.

At the same time, there are specific comments to this article:

Lines 57, 105, 116: the formula of water glass (Na3SiO4) is given. In this formula, the balance of valences does not converge.

RE:  Thank you very much for your help. We change in all places.

Line 97: 0.70 mm diameter of MiniBars™. At the same time, the length should be specified.

RE:  Thank you very much we improved this section.

Line 100: Fly-ash samples were characterized in work [6] using a spectrofluorometer (JASCO FP-6500, Japan) for the composition and the size distribution, which was 0.103 μm at Dv50. It is also necessary to indicate the specific surface area of fly ash.

RE:  Unfortunately it is end of year and it is very hard to make a new investigation, because many co-authors work at reports for the projects. I tried but I can not do it. I attached the results for the size distribution used for fly ash.

Lines167-170: It is indicated that the phases that can be detected in fly ash are: quartz - SiO2 (PDF#461045), mullite - 3Al2O3·SiO2 (PDF#150776) and hematite - Fe2O3 (PDF#330664) in small quantities. In the used fly ash, the vitreous phase (spherical or irregular) exists  together with the crystalline phase [6].

It is also necessary to present the quantitative content of these crystalline phases, as well as the content of the vitreous phase. It is also necessary to give the equation for the interaction of liquid glass with fly ash components and to determine which interaction products are formed and to present their stoichiometry

RE:  I was able to contact a colleague for this investigation and this was included at acknowledgments. Regarding to “interaction products are formed and to present their stoichiometry“ I did not made this in this study and I could investigate in details in other article with more methods of investigations.

Line 183: All mechanical testing values increase (Figures 4) with addition of MiniBars™  and were more improved than Fly100.  It is necessary to provide compressive strength at cubic test for the tested samples of Fly 100.

RE:  Sorry but I can not correlate the compressive strength at cubic test for the tested samples of Fly 100. All time I compare materials tested by the same test and all my mechanical test had the references Fly 100.

Figure 5. The photographs of the section of samples of MiniBars™ FRBC: a) MiniBars12.5; b) 221 MiniBars25; c) MiniBars50 and d) MiniBars75.  This shows the uneven distribution of fiber over the area of samples. Therefore, the anisotropy of mechanical characteristics should be manifested. However, this factor is not taken into account in the article.

RE:  Thank you very much I improved this section

Reviewer 2 Report

Comments and Suggestions for Authors

The authors fabricated a kind of composite and tested the properties. Here are some comments:

1 The authors used a material called MiniBars but didn’t provide enough information. Please explain more about MiniBars, such as the composition, microstructure, density, and difference with glass/basalt fibers. The authors also mentioned that “improves post-cracking mechanical properties of hardened concrete, increasing toughness, impact and increase fatigue resistance of concrete”. Please explain the mechanisms for these advantages.

2 For Figure 4,

The data should be plotted with MiniBars concentration as x and properties with y, instead of using bar charts. Notably, the concentrations are not equally distributed (i.e., there are two intervals, 12.5% and 25%), so the linear regression with a bar chart is meaningless.

3 For Figure 5,

The authors mentioned that “These images show an idea how samples were prepared and the MiniBars™ distribution was made in the rectangular mold.” The authors should provide a clear explanation or discussion for the figure (e.g., how they are distributed and why), instead of just saying that “this is how the sample looks like”.

4 For Figure 6,

Similar for this figure, the authors only mentioned that “Figure 6 show how MiniBars™ stop the crack propagation in the geopolymer matrix from MiniBars™ FRBC under load force and materials had elastic behavior (Figure 6d)”, but didn’t explain clearly how the cracks were propagated.

Comments on the Quality of English Language

The quality of English is good

Author Response

Dear reviewer 2,

I would like to thank you very much for the time spent to read our paper.

We try to improve the article by making some changes in the text. The new changes in the article were colored turqoise highlight.

Thank you very much for all your help. Please see below my answer to your suggestions:

RE:  Thank you for you’re suggestions, we made all the changes.

Comments and Suggestions for Authors

The authors fabricated a kind of composite and tested the properties. Here are some comments:

1 The authors used a material called MiniBars but didn’t provide enough information. Please explain more about MiniBars, such as the composition, microstructure, density, and difference with glass/basalt fibers. The authors also mentioned that “improves post-cracking mechanical properties of hardened concrete, increasing toughness, impact and increase fatigue resistance of concrete”. Please explain the mechanisms for these advantages.

RE:  Basalt MiniBars™ are CE marked and commercial materials, received from ReforceTech AS, Røyken, Norway. In this study I try to evaluate new materials obtained from MiniBars™ basalt fiber reinforced geopolymer composites (MiniBars™ FRBC). Unfortunately, my study was not to evaluate the composition, microstructure, density, and difference with glass vs. basalt fibers. For all I have mentioned I gave the references I don t have more information. In the discussion part I have explained using my experience in this field vs. the tests used in this article. More information are not available.

„MiniBars™ is a high-performance basalt fibre composite, that can provide advantages of concrete: improves post-cracking mechanical properties of hardened concrete, increasing toughness, impact and increase fatigue resistance of concrete and does not corrode [10].

  1. MiniBars™. High performance composite macrofiber for concrete reinforcement. Accessed November 21, 2023. https://reforcetech.com/technology/minibars/”

2 For Figure 4,

The data should be plotted with MiniBars concentration as x and properties with y, instead of using bar charts. Notably, the concentrations are not equally distributed (i.e., there are two intervals, 12.5% and 25%), so the linear regression with a bar chart is meaningless.

RE:  Thank you for your remark. Sure I calculate the linear regression between the points and is the same even I chose this type of graphs or points. Remove line and equations, regression from graphs and I keep regression value it only in discusions.

3 For Figure 5,

The authors mentioned that “These images show an idea how samples were prepared and the MiniBars™ distribution was made in the rectangular mold.” The authors should provide a clear explanation or discussion for the figure (e.g., how they are distributed and why), instead of just saying that “this is how the sample looks like”.

RE:  Thank you for suggestion. We improved this section.

4 For Figure 6,

Similar for this figure, the authors only mentioned that “Figure 6 show how MiniBars™ stop the crack propagation in the geopolymer matrix from MiniBars™ FRBC under load force and materials had elastic behavior (Figure 6d)”, but didn’t explain clearly how the cracks were propagated.

RE:  Thank you for suggestion. We improved this section.

Round 2

Reviewer 2 Report

Comments and Suggestions for Authors

The manuscript has been revised based on the reviewers' comments. I think it can be accepted.